# Effect of sustained virologic response on liver-related mortality among individuals living with hepatitis C by treatment era: A population-based retrospective cohort study

Aysegul Erman[1,2], Karl Everett[2], William W. L. Wong[1,2,3], Farinaz Forouzannia[3], Christina Greenaway[4], Naveed Janjua[5], Jeffrey C. Kwong[2,6,7], Beate Sander[1,2,7,8]*

1 Toronto Health Economics and Technology Assessment Collaborative (THETA), Toronto General Hospital Research Institute, University Health Network, Toronto, ON, Canada, 2 ICES, Toronto, ON, Canada, 3 School of Pharmacy, University of Waterloo, Kitchener, ON, Canada, 4 Division of Infectious Diseases, Jewish General Hospital, McGill University, Montreal, QC, Canada, 5 British Columbia Centre for Disease Control (BCDC), Vancouver, BC, Canada, 6 Department of Family and Community Medicine, Temerty Faculty of Medicine, University of Toronto, Toronto, ON, Canada, 7 Public Health Ontario, Toronto, ON, Canada, 8 Institute of Health Policy, Management, and Evaluation, University of Toronto, Toronto, ON, Canada

* beate.sander@uhn.ca

## Abstract

### Purpose

Sustained virologic response (SVR) is a validated surrogate marker for successful hepatitis C virus (HCV) treatment. Historically, interferon-based therapies, the standard of care for decades, offered only limited efficacy with respect to SVR. The recent introduction of highly effective direct-acting antivirals (DAAs) revolutionised treatment, expanding treatment eligibility among individuals with advanced liver disease (ALD) and drug/alcohol-related substance use disorder. Given these clinical policy shifts, we assessed the real-world impact of SVR on liver-related death for these key clinical groups for whom treatment had previously been less feasible.

### Methods

We conducted a population-based, cohort study of Ontario residents with HCV viremia between January 1st, 1999, and December 31st, 2018, with follow-up to May 31st, 2021 (N = 73,411) and used cause-specific hazard models to explore the association between SVR and liver-related death.

### Results

SVR was associated with a significant reduction in liver-related deaths (adjusted hazard ratio [aHR]: 0.22, 95%CI: 0.20–0.24). This benefit was consistent across all levels of liver disease severity, including individuals with (aHR: 0.11, 95%CI: 0.06–0.18) and

**Data availability statement:** Data for this study are owned by ICES, a third-party organization. Data is held securely in coded form at ICES. While legal data sharing agreements between ICES and data providers (e.g., healthcare organizations and government) prohibit ICES from making the dataset publicly available, access may be granted to those who meet pre-specified criteria for confidential access, available at https://urldefense.com/v3/__http://www.ices.on.ca/DAS__;!!CjcC7IQ!OMGiQJQ6IAkA_fED8jwBmSJ4rxFT44jUr_q1uXyrfGwCeuOQMC-q6XyDFJxkusSTQn070zzxvkucPpPQ2iRGBt-M4$[ices[.]on[.]ca] (email: das@ices.on.ca).

**Funding:** This study was supported by the ICES, which is funded by an annual grant from the Ontario Ministry of Health (MOH) and the Ministry of Long-Term Care (MLTC). A. Erman is supported by a postdoctoral fellowship from the Canadian Institute of Health Research (CIHR) [FRN:201910MFE-430962-169632]. This research was supported, in part, by a Canada Research Chair in Economics of Infectious Diseases held by Beate Sander [CRC-950-232429]. This study also received funding from CIHR grant [PJT-156066]. The analyses, conclusions, opinions, and statements expressed herein are solely those of the authors and do not reflect those of the funding or data sources; no endorsement is intended or should be inferred.

**Competing interests:** Dr. W. Wong reports a grant from Canadian Liver Foundation, outside the submitted work.

without (aHR: 0.13, 95%CI: 0.10–0.17) cirrhosis, individuals with ALD (aHR: 0.24, 95%CI: 0.22–0.27) as well as among individuals with (aHR: 0.24, 95%CI: 0.21–0.27) and without (aHR: 0.21, 95%CI: 0.18–0.24) substance use disorder.

## Conclusions

This study demonstrates the real-world impact of SVR on liver-related mortality and highlights the value of early treatment and continued support for populations who are marginalised.

---

## Introduction

Chronic hepatitis C (CHC) is associated with major morbidity and mortality. Over time, the chronic persistence of hepatitis C virus (HCV) can cause progressive liver damage, ranging from hepatic fibrosis to cirrhosis, and end-stage liver disease, often requiring liver transplantation or resulting in liver-related mortality [1–3].

Sustained virologic response (SVR) is a clinically relevant surrogate endpoint in HCV treatment, defined as the absence of detectable HCV RNA in the blood 12 or 24 weeks after completing antiviral therapy. Indeed, the achievement of SVR has been shown to substantially reduce mortality for individuals living with hepatitis C [4,5]. However, for decades the standard-of-care for HCV consisted of interferon-based therapy, offering only limited efficacy with respect to SVR [5,6]. Moreover, interferon-based therapies were of limited use for individuals with a history of substance use and mental illness due to poor tolerability and neuropsychiatric side effects [6]. Similarly, tolerability was also a concern for those with advanced liver disease (ALD) [7]. In fact, interferon-based regiments were contraindicated for those with decompensated cirrhosis (DC) and demonstrated only limited efficacy in treating recurrent infection among immunosuppressed liver-transplant recipients [7,8].

Since the introduction of highly effective direct-acting antivirals (DAAs) in 2013, clinical policy guidelines for CHC have been rapidly changing. DAAs are significantly more effective and better tolerated, enabling treatment for individuals who were previously ineligible, including those with ALD and comorbid conditions such as substance use. [9–12]. Moreover, compared to earlier therapies, DAAs demonstrate superior outcomes among post-transplant individuals [8,13], and eliminate the need for genotype-specific regimens due to their pan-genotypic activity [14]. Finally, the high rates of SVR exhibited by DAAs also offer an opportunity to eliminate HCV as a public health concern and meet the WHO target of a 65% reduction in mortality by 2030 [15]. However, despite rapid shifts in clinical practice and public health policy, real-world impact of SVR on long-term outcome such as liver-related death in the DAA era, specifically among higher-risk populations is lacking.

Our primary objective was to assess the real-world impact of SVR on liver-related death in the DAA and pre-DAA treatment eras among important clinical population such as those with advanced liver disease and substance use and our secondary

objective was to explore changes in predictors of liver deaths by treatment era from a population-based cohort of individuals living with hepatitis C in the province of Ontario, Canada's largest jurisdiction.

## Materials and methods

### Study population and setting

In Ontario, Canada, confirmation, and testing of reportable diseases including HCV and hepatitis B virus (HBV) typically occurs at Public Health Ontario (PHO) laboratories. We performed a retrospective cohort study of Ontario residents undergoing confirmatory testing for HCV viremia at PHO laboratories between January 1st, 1999, and December 31st, 2018 (N = 79,899). After the exclusion of individuals (N = 6,488) whose laboratory testing records could not be linked to demographic data of all Ontario residents held at ICES (formerly known as the Institute for Clinical Evaluative Sciences), our study cohort included 73,411 unique individuals with confirmed HCV viremia. We followed the study cohort up to May 31st, 2021, to identify total and liver-related mortality.

### Treatment era

The study cohort consisting of individuals testing positive for HCV RNA in Ontario between January 1999 and December 31st, 2018, were further classified by treatment era into 1) individuals testing HCV RNA positive during the pre-DAA era (Jan 1999- Dec 2013) and followed up until the end of the pre-DAA era (N = 54,854); and 2) individuals testing HCV RNA positive during the DAA era (Jan 2014- Dec 2018) and followed up until May 31st, 2021 (N = 18,557).

### Data sources

This study was approved by the University Health Network Research Ethics Board in Toronto, Canada. We linked health administrative, clinical, and demographic data held at ICES datasets using unique identifiers and analysed the data at ICES. Data for this study were accessed between 25/11/2020–30/10/2022. ICES is an independent, non-profit research institute whose legal status under Ontario's health information privacy law authorizes the ICES to collect personal health information, without consent, for health system evaluation, monitoring and improvement.

All data related to HBV and HCV laboratory testing were collected from PHO laboratory dataset. Data on HCV antiviral dispensation were sourced from Ontario Drug Benefit (ODB) using drug identification numbers (DIN) listed in S1 Table. For those with a record of HCV antiviral drug dispensation from public drug plans, SVR was defined as a negative HCV RNA test result at least 12 weeks after the dispensation of the last HCV antiviral drug; those without a record of HCV antiviral drug dispensation from public sources but who had a final negative HCV RNA test result that took place after a positive HCV RNA were presumed to have achieved SVR through access to treatment via private insurance schemes external to ODB. The Ontario Marginalisation Index Database (ONMARG) was used to access information on social marginalisation (e.g., neighborhood residential instability, material deprivation and ethnic concentration quintiles). We used Immigration, Refugees and Citizenship Canada (IRCC) permanent Resident Database, which holds data individuals who have been granted permanent resident status in Canada since 1985 to determine immigration status. Demographic data including birth year, gender, age, rurality, and neighborhood income quintile were accessed using the Registered Persons Database (RPD). Clinical information on alcohol and/or drug-related substance use disorder, prior diagnosis of cirrhosis, DC, hepatocellular carcinoma (HCC), liver transplant, HIV status, cause of death and comorbidities (Aggregated Diagnosis Groups [ADG] comorbidity classification scheme) were collected from the National Ambulatory Care Reporting System (NACRS), Ontario Health Insurance Program (OHIP), Canadian Institute for Health Information Discharge Abstract Database (DAD), Ontario Mental Health Reporting System (OMHRS), Ontario Cancer Registry (OCR) and Office of the Registrar General Death registry (ORGD) datasets, using the relevant diagnostic and procedural codes listed in S2 and S3 Tables and death codes listed in S4 Table. For

comorbidities, we used the Johns Hopkins Adjusted Clinical Group® (ACG) system to determine the number ADGs using a one-year look-back period preceding HCV RNA positivity [16]. Baseline characteristics of the study cohort were assembled at time of HCV RNA positivity.

## Statistical analysis

To identify factors associated with liver-related mortality, we used cause-specific hazard models accounting for non-liver related death as a competing risk [8]. For the cause-specific hazard models, individuals were followed from index date (first date of HCV RNA positivity on record) to either liver-related death, or until other death or censoring on (December 31$^{st}$, 2013, for pre-DAA era and on May 31$^{st}$, 2021, for the DAA-era cohorts) with the competing event treated as censored [9]. The hazard models were adjusted for potential confounders including age at diagnosis, sex, rurality, birth year, comorbidities, neighborhood income, immigrant status, socioeconomic markers, HIV or HBV coinfection, HCV genotype, alcohol and drug-related substance use disorder, history of liver transplant, liver disease severity.

Stratified analysis was conducted to account for clinical policy changes with respect to the expanded treatment of individuals with advanced liver disease and substance use over the two treatment eras, and to explore the moderating effects of liver disease severity and substance-use on the relationship between SVR and liver-related mortality. The analysis was stratified by the liver disease severity at time of HCV diagnosis into 1) no cirrhosis, 2) compensated cirrhosis, and 2) advanced liver disease (HCC, DC or liver transplant) and by the presence or absence of alcohol and/or drug-related substance-use disorder on record.

The predicted incidence rates of liver-related mortality was estimated using Poisson regression stratified by age at infection and by SVR status. For statistical models, we used complete case analysis by excluding observations with missing data. All statistical analysis were performed with SAS Enterprise Guide 7.15 (SAS Institute, Inc.).

# Results

## Characteristics of the study cohort

The study cohort consisted of 73,411 individuals with a positive HCV RNA test in the province of Ontario who were followed up over a median follow-up time of 10 years (IQR: 6–15 years) (Table 1). On average, individuals were 45 years of age at the time of HCV RNA positivity, 65% were male, 12% were immigrants, 1.2% were coinfected with HIV, 0.6% with HBV and 50% had substance use disorder during the study time frame. More than half of the study cohort had markers of social marginalisation such as lower income levels, higher levels of residential instability and material deprivation. With respect to liver disease severity, the majority (78%) had no evidence of cirrhosis, 7% had compensated cirrhosis, 10% had DC and 4% had HCC at diagnosis. In total 47% of the cohort received treatment of those 80.5% achieved SVR.

## Characteristics of the study cohort by treatment era

The majority (75%) of the study cohort were diagnosed in the pre-DAA era (Table 1). The median follow-up time was 7 years (IQR: 4–11 years) for individuals diagnosed and followed up within the pre-DAA era, and 5 years (IQR: 3–6 years) for those diagnosed in the DAA era. On average, individuals diagnosed in the DAA era were younger; most were born after 1965 (55% vs. 29% in the pre-DAA era).

DAA-era cases also had higher levels of substance use disorder (60% vs. 46%), were less likely to be immigrants (9% vs. 13%) and more likely to reside in rural areas (13% vs. 9%). Individuals identified during the DAA era tended to have milder liver disease (86% vs. 76%) and were less likely to be coinfected with HIV (0.8% vs. 1.4%). Lastly, SVR was also more frequent among those diagnosed (but not necessarily treated) in the DAA era vs. the pre-DAA era (83% vs. 77%).

**Table 1. Demographic characteristics of study cohort stratified by treatment era.**

| | TREATMENT ERA | | |
| --- | --- | --- | --- |
| | **Study Cohort N = 73,411** | **PRE-DAA N = 54,854** | **DAA N = 18,557** |
| **Age in years, mean (SD)** | | | |
| At time of diagnosis | 44.7 (12.9) | 45.0 (12.2) | 43.8 (14.8) |
| **Birth year, N (%)** | | | |
| <1945 | 5,513 (7.5) | 4,841 (8.8) | 672 (3.6) |
| 1945-1965 | 41,860 (57.0) | 34,204 (62.4) | 7,656 (41.3) |
| >1965 | 26,038 (35.5) | 15,809 (28.8) | 10,229 (55.1) |
| **Male sex, N (%)** | 47,756 (65.1) | 35,781 (65.2) | 11,975 (64.5) |
| **Rural, N (%)** | 7,468 (10.2) | 5,010 (9.1) | 2,458 (13.2) |
| **Neighborhood income quintile, N (%)** | | | |
| Low (q. 1–2) | 41,523 (56.6) | 30,362 (55.4) | 11,161 (60.1) |
| Medium (q. 3) | 12,510 (17.0) | 9,497 (17.3) | 3,013 (16.2) |
| High (q. 4–5) | 17,969 (24.5) | 13,930 (25.4) | 4,039 (21.8) |
| **Residential instability quintile, N (%)** | | | |
| Low (q. 1–2) | 16,904 (23.0) | 13,169 (24.0) | 3,735 (20.1) |
| Medium (q. 3) | 11,240 (15.3) | 8,487 (15.5) | 2,753 (14.8) |
| High (q. 4–5) | 42,150 (57.4) | 31,059 (56.6) | 11,091 (59.8) |
| **Material deprivation quintile, N (%)** | | | |
| Low (q. 1–2) | 17,680 (24.1) | 13,548 (24.7) | 4,132 (22.3) |
| Medium (q. 3) | 12,064 (16.4) | 9,161 (16.7) | 2,903 (15.6) |
| High (q. 4–5) | 40,550 (55.2) | 30,006 (54.7) | 10,544 (56.8) |
| **Ethnic concentration quintile, N (%)** | | | |
| Low (q. 1–2) | 25,205 (34.3) | 17,864 (32.6) | 7,341 (39.6) |
| Medium (q. 3) | 12,942 (17.6) | 9,461 (17.2) | 3,481 (18.8) |
| High (q. 4–5) | 32,147 (43.8) | 25,390 (46.3) | 6,757 (36.4) |
| **Immigrant, N (%)** | 8,843 (12.0) | 7,210 (13.1) | 1,633 (8.8) |
| **Substance use disorder (ever), N (%)** | 36,378 (49.6) | 25,221 (46.0) | 11,157 (60.1) |
| **HIV positivity, N (%)** | 900 (1.2) | 760 (1.4) | 140 (0.8) |
| **HBsAg positivity[a], N (%)** | 463 (0.6) | 358 (0.7) | 105 (0.6) |
| **Aggregated diagnosis group categories, N (%)** | | | |
| 0-3 ADG | 26,106 (35.6) | 19,157 (34.9) | 6,949 (37.4) |
| 4-7 ADG | 29,345 (40.0) | 22,351 (40.7) | 6,994 (37.7) |
| 8-10 ADG | 11,462 (15.6) | 8,643 (15.8) | 2,819 (15.2) |
| >11 ADG | 6,410 (8.7) | 4,648 (8.5) | 1,762 (9.5) |
| **Liver disease at the time of diagnosis, N (%)** | | | |
| No cirrhosis | 57,568 (78.4) | 41,590 (75.8) | 15,978 (86.1) |
| Compensated cirrhosis | 5,058 (6.9) | 4,051 (7.4) | 1,007 (5.4) |
| Decompensated cirrhosis | 7,533 (10.3) | 6,477 (11.8) | 1,056 (5.7) |
| Hepatocellular carcinoma | 3,252 (4.4) | 2,736 (5.0) | 516 (2.8) |
| **Liver transplant, N (%)** | 994 (1.4) | 782 (1.4) | 212 (1.1) |
| **HCV genotype, N (%)** | | | |
| Genotype-1 | 43,268 (58.9) | 32,780 (59.8) | 10,488 (56.5) |
| Genotype-2 | 7,391 (10.1) | 5,958 (10.9) | 1,433 (7.7) |
| Genotype-3 | 13,850 (18.9) | 9,576 (17.5) | 4,274 (23.0) |
| Genotype-4 | 1,384 (1.9) | 1,132 (2.1) | 252 (1.4) |
| Other/mixed | 1,534 (2.1) | 900 (1.6) | 634 (3.4) |

*(Continued)*

**Table 1.** (Continued)

| | TREATMENT ERA | | |
| --- | --- | --- | --- |
| | Study Cohort N = 73,411 | PRE-DAA N = 54,854 | DAA N = 18,557 |
| Treated, N (%)[b] | 34,932 (47.6) | 19,047 (34.7) | 7,571 (40.8) |
| SVR, N (%)[b] | 28,110 (80.5) | 14,732 (77.3) | 6,269 (82.8) |

Demographic characteristics of the full study cohort at the time of HCV RNA positivity and characteristics of individuals diagnosed (*but not necessarily treated*) during the pre-DAA era (Jan 1999- Dec 2013) and DAA era (Jan 2014- Dec 2018). Frequencies are calculated after exclusion of missing values. [a]HBV diagnosis is based on hepatitis B surface antigen (HBsAg) reactivity. [b]The numbers in the pre-DAA and DAA columns may not sum as expected due to differences in follow-up time. Patients diagnosed in the pre-DAA era (Jan 1999–Dec 2013) were only followed until the end of that era, so any treatment they received after that point is not captured in the pre-DAA column. However, the full study cohort includes follow-up through the entire study period; therefore, if individuals from the pre-DAA cohort received treatment during the DAA era, it is now reflected Study Cohort column. *Abbreviations: ADG: aggregated diagnostic groups; DAA: direct-acting antiviral; HBsAg: hepatitis B surface antigen; HBV: hepatitis B virus; HCV: hepatitis C virus; HIV: human immunodeficiency virus; N: number of observations; q: quintile; RNA: ribonucleic acid; SD: standard deviation; SVR: sustained virologic response.*

## Characteristics of the study cohort by liver disease severity

The study cohort consisted of 57,568 individuals without any record of cirrhosis, 5,058 individuals with compensated cirrhosis, and 10,785 with advanced liver disease (DC and/or HCC) at the time of HCV diagnosis (S5 Table). When compared to individuals presenting with mild liver disease those with more advanced liver disease tended to be older at diagnosis (52 vs. 43 years), male (72% vs. 63%), and less likely to have substance use disorder (45% vs. 49%). Finally, SVR was 81% for individuals without cirrhosis, 83% for those with compensated cirrhosis, and 75% for those with more advanced liver disease.

## Characteristics of the study cohort by liver disease severity and treatment era

With respect to treatment era, 25% of individuals with mild liver disease, 20% of individuals with compensated cirrhosis, and 15% of individuals with advanced liver disease were diagnosed in the DAA era (S5 Table). For all levels of liver disease, individuals diagnosed in the DAA era were younger, more likely to reside in lower income neighbourhoods and more likely to have a history of substance use but were less likely to be immigrant. In general, the DAA-era improvement in treatment initiation was more pronounced for those with compensated cirrhosis (50% vs. 41%) and with advanced liver disease (44% vs. 35%). There were also improvements in SVR in the DAA era among individuals presenting with compensated cirrhosis (96% vs. 77%), and with advanced liver disease (89% vs. 59%) vs. pre-DAA era. Thus, in the treatment era analyses, outcome and exposure data for subjects in the pre-DAA era were censored after 2018 (at the start of the DAA era).

## Characteristics of the study cohort by substance use

The study cohort consisted of 37,033 individuals without and 36,378 individuals with a history of drug and/or alcohol-related substance use disorder (S6 Table). Compared to individuals without substance use, those with substance use tended to be younger at diagnosis (41 vs. 48 years), male (68% vs. 62%), more likely to be HIV-coinfected (1.6% vs. 0.9%) and tended to have markers of social marginalisation including lower income (63% vs. 52%), higher residential instability (68% vs. 52%) and greater material deprivation (63% vs. 52%). Individuals with substance use disorder were also much less likely to be immigrants (3% vs. 21%) but were more likely to present with DC at diagnosis (12% vs. 8%). Treatment rates were also lower among those with substance use disorder (43% vs. 52%), as were SVR levels (73% vs. 87%).

## Characteristics of the study cohort by substance and treatment era

In terms of treatment era, 31% of individuals with substance use disorder and 20% of those without were diagnosed in the DAA era (S6 Table). Individuals with substance use who were diagnosed during the DAA era were slightly younger vs. the pre-DAA era (39 vs. 42 years). There were improvements in treatment initiation during the DAA era for both Individuals with substance use (36% vs. 29%) and those without (47% vs. 40%). SVR rates also improved during the DAA era but only among individuals without substance use disorder (96% vs. 81%).

## Incidence of liver-related death in the study cohort

Incidence of liver-related death in the study cohort by SVR status and age at diagnosis is illustrated in Fig 1 for clinical populations assessed. The incidence of liver-related, non-liver related and all cause deaths for all subgroups by SVR status is presented in S7 Table. Overall, the incidence of liver-related death in the study population was 4 per 1,000 person-years (PY) [5.9/1,000 PY among those without SVR and 1.5/1,000 PY among those with SVR]. The incidence of liver-related death increased noticeably with progressive liver disease from 1.1/1,000 PY among those without cirrhosis [1.7/1,000 PY without and 0.2/1,000 PY with SVR] to 22/1,000 PY among those presenting with advanced liver disease [34.7/1,000 PY without and 9/1,000 PY with SVR]. On average, incidence of liver-death was 4.8/1,000 PY for individuals with a history substance use [6.6/1,000 PY without and 1.8/1,000 PY with SVR] and 3.3/1,000 PY among those without substance use [5.3/1,000 PY without and 1.2/1,000 PY with SVR].

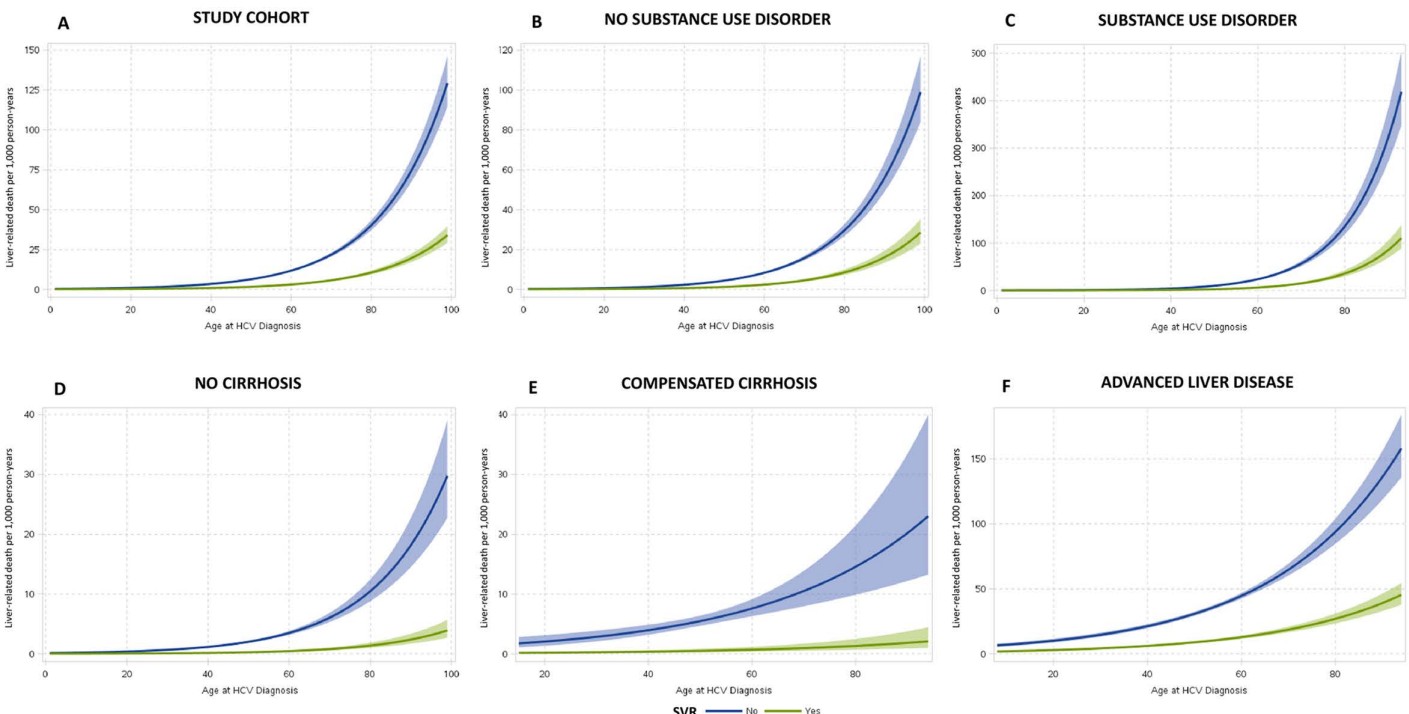

**Fig 1. Predicted incidence of liver-related death in individuals diagnosed with hepatitis C by SVR status and age at diagnosis.** Figure displays the predicted incidence rate of liver-related death per 1,000 person years by SVR status and age at diagnosis for (A) the total study cohort; (B) those without drug and/or alcohol related substance use disorder; (C) those with drug and/or alcohol related substance use disorder; and by liver disease severity for (D) those without cirrhosis; (E) those with compensated cirrhosis; and (F) those with advanced liver disease. Predicted incidence rates were estimated from Poisson regression. *Abbreviations: HCV: hepatitis C virus; SVR: sustained virologic response.*

## Predictors of liver-related death

The predictors of liver-related mortality for the study cohort are displayed in Table 2. After adjustment for covariates in the cause-specific hazard model, achievement of SVR was associated with a significant reduction in the rate of liver-related mortality (adjusted hazard ratio (aHR) 0.22, 95%CI: 0.20–0.24 vs. no SVR). Additionally, covariates associated with the timing of liver-related death included age (aHR 1.06, 95%CI: 1.06–1.07), liver disease severity at the time of diagnosis (aHR 2.18, 95%CI: 1.85–2.58 for compensated cirrhosis, aHR 6.34, 95%CI: 6.67–7.08 for DC and, aHR 26.2, 95%CI: 23.9–28.9 for HCC vs. no cirrhosis), sex (aHR 0.75, 95%CI: 0.69–0.82 for female vs. male), immigration status (aHR 0.88, 95%CI: 0.78–0.99 for immigrants vs. long-term resident), birth year (aHR 0.58, 95%CI: 0.51–0.66 for those born before 1945, and aHR 0.37, 95%CI: 0.29–0.48 for those born after 1965 vs. individuals born between 1945 and 1965), presence of substance use disorder (aHR 1.56, 95%CI: 1.44–1.68), and viral genotype (aHR 0.84, 95%CI: 0.74–0.95 for viral genotype-2 and aHR 0.78, 95%CI: 0.62–0.97 for viral genotype-4 vs. genotype-1).

Moreover, there were important differences with respect to predictors of liver-related death by treatment era. Upon stratification and follow-up by treatment era, several factors including sex, viral genotype, ethnic concentration, and the presence of compensated cirrhosis at diagnosis no longer displayed an association with liver-related death in the DAA era as was observed in the earlier treatment era. However, material deprivation was identified as a predictor (aHR 1.61, 95%CI: 1.11–2.34) in the DAA era.

## Predictors of liver-related death by liver disease severity

In the stratified analysis, SVR was again associated with a significant lower rate of liver-related death across all levels of liver disease for individuals presenting with no cirrhosis (aHR 0.13, 95%CI: 0.10–0.17), those with compensated cirrhosis (aHR 0.11, 95%CI: 0.06–0.18) and those presenting with advanced liver disease at the time of HCV diagnosis (aHR 0.24, 95%CI: 0.22–0.27) (Table 3). Covariates consistently associated with liver-related death across all stages of liver disease included age at diagnosis, birth year, sex, substance use disorder, comorbidities, and achievement of SVR. Higher income was associated with a lower rate of liver-related death only for those presenting with no cirrhosis at diagnosis (aHR 0.59, 95%CI: 0.451–0.78 vs. low level).

Immigrant status was associated with a lower rate of liver-related death (aHR 0.86, 95%CI: 0.75–0.99 vs. long-term resident), and HBV coinfection was associated with a higher incidence of liver-related death (aHR 1.65, 95%CI: 1.01–2.72) for those presenting with advanced liver disease.

Additionally, there were several notable differences in terms of predictors of liver-related death by liver disease levels across treatment eras. In the DAA era, substance use was no longer a predictor of liver-related mortality for individuals with more advanced liver disease while material deprivation was found to be a predictor of liver-related death among individuals presenting with advanced liver disease. Although, the DAA-era analysis of liver-related death among the non-cirrhotic and more specifically compensated cirrhosis groups are likely to be underpowered given low event rates.

## Predictors of liver-related death by substance use

The effect of SVR on liver-related mortality was similar regardless of individual's substance use history resulting in significantly lower rates of liver-related death for individuals with substance use (aHR 0.24, 95%CI: 0.21–0.27) as well as for individuals without (aHR 0.21, 95%CI: 0.18–0.24) (Table 4). The covariates associated with liver-related death for those with and without substance use included age at diagnosis, birth year, sex, liver disease severity at diagnosis, genotype, and achievement of SVR. However, HBV coinfection was a predictor of liver-related death for individuals with substance use (aHR 2.76, 95%CI: 1.65–4.59).

There were also differences in predictors by treatment era. For instance, in the DAA era, sex and viral genotype were no longer a significant predictor of liver-related death for either group. Similarly, residential instability and material

**Table 2.** Predictors of liver-related mortality.

| | Study Cohort N=69,558 | TREATMENT ERA | |
| --- | --- | --- | --- |
| | | PRE-DAA N=51,987 | DAA N=17,563 |
| **Covariates** | Adjusted HR (95% CI) | | |
| **Age (years)** | | | |
| At time of diagnosis | **1.06 (1.06-1.07)** | **1.09 (1.09-1.10)** | **1.02 (1.00-1.04)** |
| **Birth year** | | | |
| 1945-1965 [ref.] | – | – | – |
| <1945 | **0.58 (0.51-0.66)** | **0.32 (0.27-0.37)** | 0.85 (0.54-1.32) |
| >1965 | **0.37 (0.29-0.48)** | **0.28 (0.14-0.55)** | **0.22 (0.12-0.40)** |
| **Sex** | | | |
| Male [ref.] | | – | – |
| Female | **0.75 (0.69-0.82)** | **0.73 (0.67-0.80)** | 0.89 (0.67-1.18) |
| **Rurality** | | | |
| Urban [ref.] | – | – | – |
| Rural | 0.96 (0.84-1.09) | 1.03 (0.89-1.18) | 0.78 (0.51-1.18) |
| **Neighborhood income quintile** | | | |
| Low (q. 1–2) [ref.] | – | – | – |
| Medium (q. 3) | 0.95 (0.85-1.06) | 0.91 (0.81-1.02) | 1.12 (0.77-1.63) |
| High (q. 4–5) | 0.89 (0.79-1.00) | 0.88 (0.78-1.00) | 1.09 (0.77-1.53) |
| **Residential instability quintile** | | | |
| Low (q. 1–2) [ref.] | – | – | – |
| Medium (q. 3) | 0.95 (0.85-1.07) | 0.93 (0.83-1.05) | 1.30 (0.84-2.01) |
| High (q. 4–5) | 0.97 (0.88-1.07) | 0.95 (0.86-1.05) | 1.03 (0.68-1.55) |
| **Material deprivation quintile** | | | |
| Low (q. 1–2) [ref.] | – | – | – |
| Medium (q. 3) | 0.95 (0.85-1.07) | 0.97 (0.86-1.09) | 1.42 (0.98-2.05) |
| High (q. 4–5) | 1.03 (0.92-1.16) | 1.08 (0.96-1.22) | **1.61 (1.11-2.34)** |
| **Ethnic concentration quintile** | | | |
| Low (q. 1–2) [ref.] | – | – | – |
| Medium (q. 3) | 0.99 (0.89-1.10) | 0.97 (0.87-1.08) | 1.16 (0.85-1.60) |
| High (q. 4–5) | **0.91 (0.83-0.99)** | **0.90 (0.82-0.99)** | 1.01 (0.76-1.33) |
| **Immigrant status** | | | |
| Long-term resident [ref.] | – | – | – |
| Immigrant | **0.88 (0.78-0.99)** | 1.04 (0.92-1.19) | 0.87 (0.56-1.35) |
| **Substance use disorder** | | | |
| No [ref.] | – | – | – |
| Yes | **1.56 (1.44-1.68)** | **1.69 (1.55-1.84)** | **1.31 (1.01-1.69)** |
| **HIV positivity** | | | |
| No [ref.] | – | – | – |
| Yes | 0.90 (0.61-1.34) | 0.84 (0.56-1.26) | 0.97 (0.13-6.98) |
| **HBsAg positivity** | | | |
| No [ref.] | – | – | – |
| Yes | **1.57 (1.00-2.47)** | 1.36 (0.83-2.22) | 2.40 (0.75-7.68) |
| **Total number of ADGs (N.)** | **1.01 (1.00-1.02)** | 1.00 (0.99-1.01) | 1.01 (0.97-1.05) |
| **Liver disease at the time of diagnosis** | | | |
| No cirrhosis [ref.] | – | – | – |

*(Continued)*

**Table 2.** (Continued)

| | Study Cohort N = 69,558 | TREATMENT ERA | |
|---|---|---|---|
| | | PRE-DAA N = 51,987 | DAA N = 17,563 |
| Compensated cirrhosis | **2.18 (1.85-2.58)** | **1.95 (1.64-2.31)** | 1.55 (0.70-3.44) |
| Decompensated cirrhosis | **6.34 (5.67-7.08)** | **4.56 (4.06-5.11)** | **9.15 (5.98-14.0)** |
| Hepatocellular carcinoma | **26.2 (23.9-28.9)** | **16.8 (15.2-18.5)** | **83.5 (59.5-117)** |
| **Liver transplant** | | | |
| No transplant [ref.] | – | – | – |
| Transplant | **1.76 (1.53-2.04)** | **1.68 (1.43-1.97)** | **2.21 (1.58-3.09)** |
| **Viral genotype** | | | |
| Genotype-1 [ref.] | – | – | – |
| Genotype-2 | **0.84 (0.74-0.95)** | **0.83 (0.73-0.95)** | 1.13 (0.78-1.64) |
| Genotype-3 | 1.07 (0.97-1.18) | **1.22 (1.10-1.36)** | 0.82 (0.60-1.10) |
| Genotype-4 | **0.78 (0.62-0.97)** | **0.75 (0.59-0.95)** | 1.06 (0.50-2.27) |
| Other/mixed | 0.94 (0.73-1.20) | 0.81 (0.61-1.09) | 0.76 (0.47-1.24) |
| **Achievement of SVR** | | | |
| No [ref.] | – | – | – |
| Yes | **0.22 (0.20-0.24)** | **0.28 (0.25-0.31)** | **0.26 (0.19-0.35)** |
| **NUMBER OF EVENTS** | 3,116 | 2,836 | 330 |

Predictors of liver-related death following HCV RNA positivity. Observations with missing covariates were excluded. Adjusted hazard ratios (HR) were determined using multivariable cause-specific hazard models. Bold values indicate statistical significance at p < 0.05. *Abbreviations: ADG: aggregated diagnostic groups; CI: confidence interval; DAA: direct-acting antiviral; HBsAg: hepatitis B surface* antigen*; HCV: hepatitis C virus; HIV: Human immuno-deficiency virus; HR: hazard ratio; N: number of observations; q: quintile; ref: reference; RNA: ribonucleic acid; SVR: sustained virologic response.*

deprivation were no longer predictors of liver-related death for individuals without substance use, and HBV-coinfection was no longer a predictor of liver-related death for those with substance use in the DAA era.

## Discussion

There have been major shifts in clinical policy guidelines for CHC following the introduction of DAAs resulting in expanded treatment access to individuals for whom HCV treatment had previously been less feasible. Despite rapid changes, real-world effects of SVR and other important predictors of liver-related death has not been assessed. The current study investigates real-world association between SVR and liver-related mortality across important clinical subgroups of individuals living with hepatitis C and explores the predictors of liver-related death in the DAA and pre-DAA treatment eras using a population-based cohort in Canada.

We found SVR to significantly reduce liver-related death independent of other covariates and across all clinical sub-populations assessed. These findings were largely congruent with a previous U.S study by Backus *et al* whereby achievement of SVR was independently associated with lower rate of all-cause death among those with advanced liver disease (defined by FIB-4 score over 3.25) receiving DAA treatment (HR 0.26, 95%CI: 0.22–0.31 vs. non SVR) [17] and a pre-DAA era multicenter study by *Van der Meer et al* among individual with advanced hepatic fibrosis or cirrhosis (HR 0.26, 95%CI: 0.14–0.49 vs. non SVR) [5]. Moreover, our findings were also consistent with more generalizable population-level real-world studies, including a DAA-era UK study which found that achievement of SVR among individuals with compensated cirrhosis to be associated with lower rates of liver-related death (HR 0.13, 95%CI: 0.05–0.34 vs. non SVR) and all-cause mortality (HR: 0.30, 95%CI: 0.12–0.76 vs. non SVR) [18]. Finally, a recent Canadian study from British Colombia also found SVR to be associated with significant reduction in liver-related deaths (HR 0.22, 95%CI: 0.18–0.27) as well as with

**Table 3. Predictors of liver-related mortality stratified by liver disease severity.**

| | | Treatment era | | | Treatment era | | | Treatment era | |
|---|---|---|---|---|---|---|---|---|---|
| | No cirrhosis N=54,235 | PRE-DAA N=39,173 | DAA N=15,054 | Compensated cirrhosis N=4,889 | PRE-DAA N=3,901 | DAA N=988 | Advanced liver disease N=10,434 | PRE-DAA N=8,913 | DAA N=1,521 |
| Covariates | Adjusted HR (95% CI) | | | Adjusted HR (95% CI) | | | Adjusted HR (95% CI) | | |
| **Age (years)** | | | | | | | | | |
| At time of diagnosis | **1.05 (1.04-1.06)** | **1.10 (1.09-1.12)** | 1.00 (0.96-1.04) | **1.04 (1.02-1.07)** | **1.09 (1.06-1.12)** | 0.99 (0.87-1.13) | **1.07 (1.06-1.08)** | **1.09 (1.08-1.10)** | **1.02 (1.00-1.04)** |
| **Birth year** | | | | | | | | | |
| 1945-1965 [ref.] | – | – | – | – | – | – | – | – | – |
| <1945 | **1.24 (0.90-1.71)** | **0.40 (0.27-0.57)** | 1.89 (0.50-7.10) | 1.08 (0.60-1.95) | **0.40 (0.21-0.78)** | **14.6 (0.80-265)** | **0.45 (0.39-0.53)** | **0.29 (0.24-0.35)** | **0.69 (0.42-1.13)** |
| >1965 | **0.12 (0.08-0.19)** | **0.38 (0.24-0.62)** | **0.05 (0.01-0.17)** | **0.34 (0.16-0.73)** | 1.01 (0.45-2.27) | – | 0.91 (0.67-1.25) | 1.38 (0.95-2.01) | 0.56 (0.29-1.06) |
| **Sex** | | | | | | | | | |
| Male [ref.] | – | – | – | – | – | – | – | – | – |
| Female | **0.70 (0.59-0.84)** | **0.72 (0.60-0.87)** | 0.69 (0.36-1.35) | **0.68 (0.48-0.95)** | **0.62 (0.44-0.88)** | – | **0.78 (0.70-0.86)** | **0.74 (0.66-0.82)** | 0.99 (0.73-1.36) |
| **Rurality** | | | | | | | | | |
| Urban [ref.] | – | – | – | – | – | – | – | – | – |
| Rural | 0.81 (0.59-1.10) | 0.88 (0.64-1.22) | 0.44 (0.10-1.94) | 0.94 (0.55-1.62) | 1.07 (0.63-1.84) | – | 1.00 (0.85-1.16) | 1.06 (0.90-1.25) | 0.85 (0.54-1.32) |
| **Neighborhood income quintile** | | | | | | | | | |
| Low (q. 1–2) [ref.] | – | – | – | – | – | – | – | – | – |
| Medium (q. 3) | 0.82 (0.65-1.05) | 0.80 (0.62-1.03) | 1.25 (0.54-2.88) | 0.98 (0.61-1.56) | 1.03 (0.65-1.64) | 0.15 (0.01-3.20) | 0.98 (0.87-1.12) | 0.94 (0.82-1.08) | 1.52 (1.05-2.21) |
| High (q. 4–5) | **0.59 (0.45-0.78)** | **0.57 (0.43-0.76)** | 0.50 (0.15-1.59) | 0.80 (0.47-1.36) | 0.86 (0.50-1.47) | **0.04 (0.01-0.98)** | 1.01 (0.88-1.15) | 1.00 (0.87-1.16) | 1.35 (0.88-2.05) |
| **Residential instability quintile** | | | | | | | | | |
| Low (q. 1–2) [ref.] | – | – | – | – | – | – | – | – | – |
| Medium (q. 3) | 0.84 (0.64-1.09) | 0.82 (0.63-1.08) | 0.80 (0.24-2.59) | 0.87 (0.55-1.38) | 0.92 (0.57-1.48) | 1.78 (0.10-32.1) | 0.98 (0.87-1.12) | 0.97 (0.84-1.11) | 1.08 (0.72-1.62) |
| High (q. 4–5) | 0.80 (0.64-1.00) | **0.76 (0.61-0.96)** | 1.30 (0.51-3.33) | 0.73 (0.48-1.12) | 0.93 (0.61-1.43) | 0.88 (0.06-11.9) | 1.03 (0.92-1.15) | 1.00 (0.89-1.13) | 1.04 (0.72-1.51) |
| **Material deprivation quintile** | | | | | | | | | |
| Low (q. 1–2) [ref.] | – | – | – | – | – | – | – | – | – |
| Medium (q. 3) | 0.92 (0.70-1.20) | 0.97 (0.73-1.28) | 0.76 (0.30-1.95) | 0.75 (0.46-1.24) | 0.76 (0.46-1.26) | – | 0.97 (0.85-1.10) | 0.97 (0.85-1.11) | **1.68 (1.12-2.51)** |
| High (q. 4–5) | 0.93 (0.72-1.19) | 1.05 (0.80-1.37) | 0.58 (0.23-1.45) | 0.83 (0.51-1.34) | 0.89 (0.54-1.46) | 0.11 (0.01-1.17) | 1.07 (0.93-1.22) | 1.11 (0.96-1.28) | **2.04 (1.35-3.08)** |
| **Ethnic concentration quintile** | | | | | | | | | |
| Low (q. 1–2) [ref.] | – | – | – | – | – | – | – | – | – |
| Medium (q. 3) | 0.85 (0.67-1.07) | 0.83 (0.65-1.06) | 0.86 (0.38-1.94) | **1.68 (1.14-2.48)** | **1.55 (1.04-2.32)** | 2.47 (0.30-20.7) | 0.99 (0.88-1.12) | 0.98 (0.86-1.11) | 1.22 (0.85-1.73) |
| High (q. 4–5) | 0.92 (0.76-1.11) | 0.87 (0.71-1.07) | 1.03 (0.53-2.01) | 0.86 (0.59-1.27) | 0.85 (0.57-1.27) | 0.77 (0.09-6.29) | 0.91 (0.82-1.01) | 0.91 (0.81-1.02) | 1.02 (0.75-1.40) |
| **Immigrant status** | | | | | | | | | |
| Long-term resident [ref.] | – | – | – | – | – | – | – | – | – |

*(Continued)*

| Covariates | No cirrhosis N=54,235 | Treatment era PRE-DAA N=39,173 | DAA N=15,054 | Compensated cirrhosis N=4,889 | Treatment era PRE-DAA N=3,901 | DAA N=988 | Advanced liver disease N=10,434 | Treatment era PRE-DAA N=8,913 | DAA N=1,521 |
|---|---|---|---|---|---|---|---|---|---|
| | Adjusted HR (95% CI) | | | Adjusted HR (95% CI) | | | Adjusted HR (95% CI) | | |
| Immigrant | 0.95 (0.71-1.27) | 1.03 (0.76-1.38) | 1.80 (0.57-5.66) | 1.01 (0.52-1.98) | 0.91 (0.46-1.78) | – | **0.86 (0.75-0.99)** | 1.05 (0.91-1.22) | 0.81 (0.50-1.30) |
| **Substance use disorder** | | | | | | | | | |
| No [ref.] | – | – | – | – | – | – | – | – | – |
| Yes | **3.30 (2.75-3.98)** | **3.51 (2.91-4.24)** | **5.69 (2.55-12.7)** | **4.76 (3.30-6.88)** | **2.47 (1.77-3.44)** | 0.96 (0.18-5.11) | **1.15 (1.05-1.26)** | **1.23 (1.12-1.36)** | 1.04 (0.79-1.38) |
| **HIV positivity** | | | | | | | | | |
| No [ref.] | – | – | – | – | – | – | – | – | – |
| Yes | 1.28 (0.68-2.40) | 1.31 (0.70-2.46) | – | – | – | – | 0.78 (0.47-1.30) | 0.70 (0.41-1.19) | 1.02 (0.99-1.06) |
| **HBsAg positivity** | | | | | | | | | |
| No [ref.] | – | – | – | – | – | – | – | – | – |
| Yes | 0.85 (0.21-3.40) | 1.06 (0.26-4.28) | – | 1.00 (0.14-7.26) | 1.31 (0.18-9.65) | – | **1.65 (1.01-2.72)** | 1.32 (0.76-2.30) | 2.31 (0.71-7.44) |
| **Total number of ADGs (N.)** | **1.03 (1.00-1.05)** | 0.99 (0.97-1.02) | 1.01 (0.94-1.09) | **1.05 (1.01-1.10)** | **1.07 (1.02-1.12)** | 0.85 (0.64-1.13) | 1.01 (0.99-1.02) | 0.99 (0.98-1.01) | 1.02 (0.99-1.06) |
| **Liver disease at the time of diagnosis** | | | | | | | | | |
| Decompensated cirrhosis [ref.] | – | – | – | – | – | – | – | – | – |
| Hepatocellular carcinoma | – | – | – | – | – | – | **4.17 (3.80-4.56)** | **3.67 (3.34-4.04)** | **9.94 (7.03-14.1)** |
| **Liver transplant** | | | | | | | | | |
| No transplant [ref.] | – | – | – | – | – | – | – | – | – |
| Transplant | – | – | – | | | | **1.74 (1.50-2.01)** | **1.67 (1.42-1.97)** | **2.38 (1.69-3.34)** |
| **Viral genotype** | | | | | | | | | |
| Genotype-1 [ref.] | – | – | – | – | – | – | – | – | – |
| Genotype-2 | **0.73 (0.56-0.94)** | **0.67 (0.51-0.88)** | 1.13 (0.50-2.57) | 0.61 (0.36-1.04) | 0.59 (0.34-1.02) | 0.54 (0.04-7.18) | 0.95 (0.81-1.10) | 0.97 (0.83-1.14) | 1.10 (0.72-1.67) |
| Genotype-3 | 1.12 (0.90-1.39) | **1.29 (1.03-1.63)** | 0.70 (0.31-1.60) | 1.02 (0.67-1.55) | 1.10 (0.71-1.70) | 1.22 (0.12-12.1) | 1.06 (0.94-1.19) | **1.22 (1.08-1.38)** | 0.83 (0.59-1.15) |
| Genotype-4 | 0.88 (0.50-1.55) | 0.86 (0.49-1.53) | – | 0.73 (0.17-3.07) | 0.87 (0.21-3.63) | – | **0.73 (0.57-0.93)** | **0.68 (0.53-0.89)** | 1.25 (0.57-2.74) |
| Other/mixed | 0.82 (0.44-1.55) | 0.94 (0.48-1.84) | 0.52 (0.07-3.87) | – | 0.73 (0.41-1.30) | – | 0.98 (0.75-1.29) | 0.79 (0.57-1.09) | 0.81 (0.49-1.34) |
| **Achievement of SVR** | | | | | | | | | |
| No [ref.] | – | – | – | – | – | – | – | – | – |
| Yes | **0.13 (0.10-0.17)** | **0.15 (0.12-0.20)** | **0.10 (0.03-0.33)** | **0.11 (0.06-0.18)** | **0.09 (0.05-0.17)** | 0.53 (0.10-2.69) | **0.24 (0.22-0.27)** | **0.33 (0.30-0.37)** | **0.27 (0.20-0.36)** |
| **NUMBER OF EVENTS** | 652 | 602 | 50 | 179 | 172 | 7 | 2,335 | 2,062 | 273 |

Predictors of liver-related death following HCV RNA positivity by liver disease severity. Observations with missing covariates were excluded. Adjusted hazard ratios (HR) were determined using multivariable cause-specific hazard models. Bold values indicate statistical significance at p<0.05. *Abbreviations: ADG: aggregated diagnostic groups; CI: confidence interval; DAA: direct-acting antiviral; HBsAg: hepatitis B surface* antigen*; HCV: hepatitis C virus; HIV: Human immunodeficiency virus; HR: hazard ratio; N: number of observations; q: quintile; ref: reference; RNA: ribonucleic acid; SVR: sustained virologic response*

**Table 4. Predictors of liver-related mortality stratified by substance use.**

| | | Treatment era | | | Treatment era | |
|---|---|---|---|---|---|---|
| | No substance use disorder N=35,108 | PRE-DAA N=27,987 | DAA N=7,116 | Substance use disorder N=34,450 | PRE-DAA N=24,000 | DAA N=10,447 |
| **Covariates** | **Adjusted HR (95% CI)** | | | **Adjusted HR (95% CI)** | | |
| **Age (years)** | | | | | | |
| At time of diagnosis | **1.07 (1.06-1.07)** | **1.09 (1.08-1.10)** | 1.01 (0.98-1.03) | **1.07 (1.06-1.07)** | **1.10 (1.09-1.11)** | **1.03 (1.01-1.05)** |
| **Birth year** | | | | | | |
| 1945-1965 [ref.] | – | – | – | – | – | – |
| <1945 | **0.56 (0.46-0.67)** | **0.33 (0.27-0.41)** | 0.86 (0.47-1.55) | **0.51 (0.41-0.64)** | **0.27 (0.21-0.34)** | 0.51 (0.19-1.35) |
| >1965 | **0.52 (0.33-0.82)** | 0.94 (0.58-1.53) | **0.20 (0.04-0.92)** | **0.30 (0.22-0.40)** | **0.62 (0.44-0.88)** | **0.22 (0.11-0.43)** |
| **Sex** | | | | | | |
| Male [ref.] | – | – | – | – | – | – |
| Female | **0.80 (0.71-0.89)** | **0.76 (0.67-0.86)** | 1.12 (0.76-1.66) | **0.70 (0.62-0.80)** | **0.70 (0.61-0.81)** | 0.68 (0.44-1.06) |
| **Rurality** | | | | | | |
| Urban [ref.] | – | – | – | – | – | – |
| Rural | 0.92 (0.75-1.14) | 0.98 (0.78-1.22) | 0.63 (0.32-1.24) | 0.97 (0.82-1.16) | 1.07 (0.88-1.28) | 0.91 (0.53-1.55) |
| **Neighborhood income quintile** | | | | | | |
| Low (q. 1–2) [ref.] | – | – | – | – | – | – |
| Medium (q. 3) | 0.98 (0.84-1.15) | 0.96 (0.82-1.14) | 1.81 (1.05-3.11) | 0.92 (0.79-1.08) | 0.86 (0.72-1.01) | 1.14 (0.73-1.78) |
| High (q. 4–5) | 0.88 (0.74-1.04) | 0.85 (0.71-1.02) | 1.71 (0.98-3.00) | 0.94 (0.80-1.10) | 0.93 (0.79-1.11) | 0.72 (0.40-1.31) |
| **Residential instability quintile** | | | | | | |
| Low (q. 1–2) [ref.] | – | – | – | – | – | – |
| Medium (q. 3) | 0.90 (0.77-1.05) | 0.87 (0.74-1.02) | 0.96 (0.57-1.63) | 1.00 (0.85-1.18) | 0.99 (0.84-1.18) | 1.20 (0.68-2.11) |
| High (q. 4–5) | 0.92 (0.80-1.05) | **0.84 (0.73-0.97)** | 1.70 (0.98-2.95) | 1.01 (0.88-1.16) | 1.05 (0.91-1.21) | 0.83 (0.50-1.38) |
| **Material deprivation quintile** | | | | | | |
| Low (q. 1–2) [ref.] | – | – | – | – | – | – |
| Medium (q. 3) | 1.01 (0.86-1.19) | 1.02 (0.86-1.21) | 1.24 (0.72-2.12) | 0.89 (0.76-1.05) | 0.90 (0.76-1.06) | 1.43 (0.85-2.43) |
| High (q. 4–5) | 1.12 (0.95-1.32) | **1.20 (1.01-1.42)** | 1.70 (0.98-2.95) | 0.95 (0.81-1.11) | 0.98 (0.83-1.16) | 1.59 (0.94-2.71) |
| **Ethnic concentration quintile** | | | | | | |
| Low (q. 1–2) [ref.] | – | – | – | – | – | – |
| Medium (q. 3) | 0.98 (0.84-1.16) | 1.01 (0.85-1.20) | 1.04 (0.62-1.75) | 1.00 (0.87-1.15) | 0.95 (0.82-1.10) | 1.26 (0.83-1.90) |
| High (q. 4–5) | **0.86 (0.75-0.99)** | 0.87 (0.75-1.01) | 1.01 (0.64-1.58) | 0.95 (0.84-1.07) | 0.94 (0.82-1.06) | 0.97 (0.68-1.40) |
| **Immigrant status** | | | | | | |
| Long-term resident [ref.] | – | – | – | – | – | – |
| Immigrant | 0.89 (0.77-1.03) | 1.05 (0.91-1.23) | 0.80 (0.47-1.35) | 0.83 (0.65-1.07) | 1.00 (0.77-1.30) | 1.31 (0.52-3.31) |
| **HIV positivity** | | | | | | |
| No [ref.] | – | – | – | – | – | – |
| Yes | 1.41 (0.83-2.39) | 1.33 (0.77-2.31) | 3.82 (0.50-29.2) | 0.58 (0.32-1.05) | **0.54 (0.30-0.98)** | – |
| **HBsAg positivity** | | | | | | |
| No [ref.] | – | – | – | – | – | – |
| Yes | 0.53 (0.20-1.42) | 0.99 (0.98-1.01) | 1.67 (0.22-12.6) | **2.76 (1.65-4.59)** | **2.93 (1.69-5.07)** | 2.42 (0.57-10.2) |
| **Total number of ADGs (N.)** | **1.01 (1.00-1.03)** | 0.99 (0.98-1.01) | 1.05 (0.99-1.11) | 1.01 (0.99-1.02) | 1.00 (0.99-1.01) | 1.01 (0.97-1.06) |
| **Liver disease at the time of diagnosis** | | | | | | |
| No cirrhosis [ref.] | – | – | – | – | – | – |
| Compensated cirrhosis | **1.53 (1.12-2.09)** | 1.29 (0.93-1.79) | **4.14 (1.29-13.3)** | **2.60 (2.13-3.17)** | **2.33 (1.9-2.85)** | 0.89 (0.27-2.89) |
| Decompensated cirrhosis | **10.9 (9.20-13.0)** | **7.78 (6.52-9.29)** | **24.7 (10.7-57.1)** | **4.03 (3.49-4.65)** | **2.87 (2.47-3.33)** | **5.78 (3.49-9.57)** |

*(Continued)*

**Table 4.** (Continued)

| Covariates | No substance use disorder N = 35,108 | Treatment era | | Substance use disorder N = 34,450 | Treatment era | |
| | | PRE-DAA N = 27,987 | DAA N = 7,116 | | PRE-DAA N = 24,000 | DAA N = 10,447 |
|---|---|---|---|---|---|---|
| | **Adjusted HR (95% CI)** | | | **Adjusted HR (95% CI)** | | |
| Hepatocellular carcinoma | **44.4 (38.3-51.5)** | **27.7 (23.8-32.2)** | **237 (120-471.8)** | **16.4 (14.4-18.6)** | **10.5 (9.14-11.9)** | **52.8 (34.6-80.6)** |
| **Liver transplant** | | | | | | |
| No transplant [ref.] | – | – | – | – | – | – |
| Transplant | **1.33 (1.05-1.68)** | 1.19 (0.90-1.55) | **2.10 (1.20-3.67)** | **2.04 (1.70-2.45)** | **2.05 (1.67-2.51)** | 1.48 (0.87-2.50) |
| **Viral genotype** | | | | | | |
| Genotype-1 [ref.] | – | – | – | – | – | – |
| Genotype-2 | 0.83 (0.69-1.00) | **0.81 (0.66-0.98)** | 0.92 (0.52-1.63) | 0.89 (0.75-1.06) | 0.88 (0.73-1.06) | 1.55 (0.94-2.57) |
| Genotype-3 | 1.02 (0.87-1.20) | **1.19 (1.01-1.40)** | 0.63 (0.37-1.08) | 1.11 (0.98-1.26) | **1.27 (1.11-1.45)** | 0.92 (0.63-1.33) |
| Genotype-4 | 0.85 (0.66-1.08) | 0.80 (0.62-1.03) | 1.08 (0.46-2.54) | **0.40 (0.20-0.81)** | **0.36 (0.17-0.75)** | 1.11 (0.13-9.46) |
| Other/mixed | 0.87 (0.73-1.04) | **0.69 (0.57-0.83)** | 1.02 (0.57-1.82) | 0.91 (0.76-1.09) | 1.35 (0.72-2.53) | 0.34 (0.12-1.00) |
| **Achievement of SVR** | | | | | | |
| No [ref.] | – | – | – | – | – | – |
| Yes | **0.24 (0.21-0.27)** | **0.30 (0.26-0.34)** | **0.29 (0.19-0.44)** | **0.21 (0.18-0.24)** | **0.26 (0.23-0.30)** | **0.21 (0.14-0.32)** |
| **NUMBER OF EVENTS** | 1,472 | 1,325 | 147 | 1,694 | 1,511 | 183 |

Predictors of liver-related death following HCV RNA positivity for those with or without alcohol and/or drug related substance use disorder. Observations with missing covariates were excluded from the analysis. Adjusted hazard ratios (HR) were determined using multivariable and univariate cause-specific hazard models respectively. Bold values indicate statistical significance at p < 0.05. *Abbreviations: ADG: aggregated diagnostic groups; CI: confidence interval; DAA: direct-acting antiviral; HBsAg: hepatitis B surface* antigen*; HCV: hepatitis C virus; HIV: Human immunodeficiency virus; HR: hazard ratio; N: number of observations; q: quintile; ref: reference; RNA: ribonucleic acid; SVR: sustained virologic response*

all-cause (HR 0.19, 95%CI: 0.17–0.21) and drug-related mortality (HR 0.26, 95%CI: 0.21–0.32) in broad population-based cohort [19]. However, the literature investigating the effects of SVR on mortality specifically among those with alcohol or drug-related substance use is limited. To our knowledge, our study is the first to evaluate the mortality effects of SVR among individuals with drug and/or alcohol related substance use disorder using real world population-based cohort study and to provide evidence that individuals with substance use disorder can experience significant mortality benefits with respect to liver-related death following achievement of SVR.

Moreover, we identified factors such as age at diagnosis, sex, liver disease severity at diagnosis, immigration status, birth year, substance use, HBV-coinfection viral genotype and markers of social marginalisation as independent predictors of liver-related mortality in addition to SVR. However, we also found important differences by treatment era, whereby factors such as sex, viral genotype, no longer displayed significant associations with liver-related death in the DAA era as was observed in the earlier treatment era. Given the profound mortality impact of liver disease severity, improvements in early diagnosis and timely access to effective therapies will be needed to reduce liver-related mortality in the province. Early treatment especially for individuals with cirrhosis remains important as this would prevent disease progression advanced liver disease, at which stage the probability of SVR is lower and where significant morbidity may persist following SVR [2].

Our study has certain limitations. First, individuals with substance use disorder are defined as those with a clinical diagnosis of drug and/or alcohol related substance use, thus individuals with either less severe or undisclosed substance use are likely to be misclassified. Second, the definition of SVR was based on a record of HCV antiviral dispensation based on public drug plan data. To account for individuals who may have received treatment outside of the provincial drug plan those without a record of HCV antiviral drug dispensation from public sources but who have a final

negative test result were assumed to have achieved SVR; thus, a small number of individuals with acute resolved HCV infection may be misclassified as having achieved SVR. Third, because PHO HCV testing data covers test performed between 1999–2018, the analyses lack data on those diagnosed prior to 1999 and may misclassify individuals who may have achieved SVR after 2018. Moreover, follow-up period of individuals identified in the DAA era is shorter than those diagnosed in the pre-DAA era. Additionally, while cirrhosis and DC cases identified using validated algorithms, the accuracy of these definitions depend on the sensitivity and specificity of the algorithm, and cases of cirrhosis may be underestimated. We also did not account for liver steatosis, a condition that may lead to advance liver disease after the SVR, therefore, we could not determine whether advanced liver disease was attributable to steatosis or other causes. Similarly, the sensitivity and specificity of death certificate-based codes used to identify liver disease are unknown and may vary across levels of disease severity. While it is possible that death certificates under-estimate liver-related deaths; we would expect this under-estimation to apply to both the SVR and non-SVR groups; thus the effect on the hazards ratios is expected to be minimal. Finally, it should also be recognised that while liver-related death reported in this study likely includes death caused by viral hepatitis as well as other causes such as alcohol-use, the findings nonetheless demonstrate that achievement of SVR has an important protective effect against liver-related mortality in all groups of individuals living with hepatitis C including those with and without significant liver disease and those with and without substance use.

Our study also has strengths. We investigated real-world predictors of liver-related death in the DAA and pre-DAA treatment eras among individuals living with hepatitis C using a population-based cohort in Ontario, Canada's most populous jurisdiction and estimated the incidence of liver-related death by SVR status for important clinical groups. The availability of real-world population-based data in the current study allows for generalizable findings compared to studies based on hepatology units [5,20]; or those relying on Medicare or U.S. Veteran populations [17,21]. Moreover, our study includes the most current population-wide HCV test results available in Ontario that could be linked to provincial health administrative records and adjusts for important clinical and demographic confounders. In addition, while many studies focus on all-cause death, given the ability to identify condition-specific deaths using linked administrative data, we were able to evaluate the effect of SVR on liver-related deaths [5,17,21]. Albeit our supplemental analysis also suggests that may SVR can have benefits in terms of non-liver-related deaths for all subgroups assessed (S7 Table).

## Conclusion

In conclusion, the current study focuses on the relationship between SVR and liver-related mortality, which is a clinically relevant endpoint in hepatitis C treatment, which cannot be easily assessed in clinical trials. We find that the achievement of SVR in the real-world setting has had a profound impact on reducing liver-related mortality, which was consistent across important clinical subpopulations. Thus, the findings challenge the notion of limiting DAA access for individuals with drug and/or alcohol misuse or the use of sobriety restrictions. However, the findings also demonstrate that disease severity at diagnosis remains as a major predictor of liver-related death followed by substance use disorder, highlighting the importance of early diagnosis and the value of supporting marginalised individuals in treatment access and supportive programmes such as harm reduction.

## Supporting information

**S1 Table. Drug identification numbers used to identify HCV antiviral treatment.**
(PDF)

**S2 Table. Diagnostic and procedure codes used to identify HCV-related diagnosis and comorbidities from datasets held at ICES.**
(PDF)

**S3 Table.  Detailed description of diagnostic, death and procedure-related codes.**
(PDF)

**S4 Table.  Death code used to identify liver-related death.**
(PDF)

**S5 Table.  Characteristics of study cohort stratified by liver disease severity and treatment era.**
(PDF)

**S6 Table.  Characteristics of study cohort stratified by substance use and treatment era.**
(PDF)

**S7 Table.  Incidence of clinical events in the study cohort by SVR achievement.**
(PDF)

## Acknowledgments

Parts or whole of this material are based on data and/or information compiled and provided by Immigration, Refugees and Citizenship Canada (IRCC) current to September 2020. However, the analyses, conclusions, opinions, and statements expressed in the material are those of the author(s), and not necessarily those of IRCC. We thank IQVIA Solutions Canada Inc. for use of their Drug Information File. Parts of this material are based on data and/or information compiled and provided by CIHI. However, the analyses, conclusions, opinions and statements expressed in the material are those of the author(s), and not necessarily those of CIHI. Parts of this report are based on Ontario Registrar General (ORG) information on deaths, the original source of which is Service Ontario. The views expressed therein are those of the author and do not necessarily reflect those of ORG or the Ministry of Public and Business Service Delivery. We thank the Toronto Community Health Profiles Partnership for providing access to the Ontario Marginalization Index. This document used data adapted from the Statistics Canada Postal CodeOM Conversion File, which is based on data licensed from Canada Post Corporation, and/or data adapted from the Ontario Ministry of Health Postal Code Conversion File, which contains data copied under license from ©Canada Post Corporation and Statistics Canada.

## Author contributions

**Conceptualization:** Aysegul Erman, Christina Greenaway, Naveed Janjua, Jeffrey C. Kwong, Beate Sander.

**Data curation:** Aysegul Erman, Karl Everett.

**Formal analysis:** Aysegul Erman, Karl Everett, William W. L. Wong, Farinaz Forouzannia.

**Funding acquisition:** Christina Greenaway, Naveed Janjua, Jeffrey C. Kwong, Beate Sander.

**Methodology:** Aysegul Erman, William W. L. Wong, Farinaz Forouzannia, Christina Greenaway, Naveed Janjua, Jeffrey C. Kwong, Beate Sander.

**Project administration:** Beate Sander.

**Supervision:** Jeffrey C. Kwong, Beate Sander.

**Validation:** Karl Everett, William W. L. Wong.

**Visualization:** Aysegul Erman.

**Writing – original draft:** Aysegul Erman.

**Writing – review & editing:** Aysegul Erman, Karl Everett, William W. L. Wong, Farinaz Forouzannia, Christina Greenaway, Naveed Janjua, Jeffrey C. Kwong, Beate Sander.

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
