## [Decision Letter · Decision Letter 0]

24 Jun 2025

Dear Dr. Sander,

Thank you for submitting your manuscript to PLOS ONE. After careful consideration, we feel that it has merit but does not fully meet PLOS ONE’s publication criteria as it currently stands. Therefore, we invite you to submit a revised version of the manuscript that addresses the points raised during the review process.

We look forward to receiving your revised manuscript.

Kind regards,

Yury E Khudyakov, PhD

Academic Editor

PLOS ONE

Journal Requirements:

“Dr. W. Wong reports a grant from Canadian Liver Foundation, outside the submitted work.”

**Additional Editor Comments:**

Your manuscript was reviewed by two experts in the field. Both identified significant problems in your submission. Please review the attached comments and provide point-by-point responses.

Reviewers' comments:

Reviewer's Responses to Questions

**Comments to the Author**

1. Is the manuscript technically sound, and do the data support the conclusions?

Reviewer #1: Partly

Reviewer #2: Yes

2. Has the statistical analysis been performed appropriately and rigorously?

Reviewer #1: I Don't Know

Reviewer #2: Yes

3. Have the authors made all data underlying the findings in their manuscript fully available?

Reviewer #1: Yes

Reviewer #2: Yes

4. Is the manuscript presented in an intelligible fashion and written in standard English?

Reviewer #1: Yes

Reviewer #2: Yes

Reviewer #1: This article iis covering an important topic on HCV. Unfortunatelly it is not easily readable, especially the abstract and introduction is not very attractive for the reader. I do not see much impact of this article to the future. On the other hand many known or expected facts have been confirmed.

Other comments and questions concerning specific part of the text which should be corrected or explained:

- line 40+397: Do you mean liver-related mortality?

- SVR was mentioned several times before you explained it (line 107) and the abbreviation is translated in two different ways in the article (sustained viral clearance - most cases and sustained virologic response- line 289 and 311). I don't think sustained viral clearance is the correct term because both European and American guidelines use the term sustained virologic response.

- line 98 and further: ICES - Is there any explanation of this abbreviation?

- line 156+165: I would suggest HCV RNA positivity instead of RNA diagnosis

- line 160-161: Why 5 % of patients had HCC at diagnosis if almost 80 % did not have cirrhosis? How was the cirrhosis diagnosis made?

- line 162: Why only 47 % recieved HCV treatment?

- line 385+404: Do you mean liver-related death?

- line 402: Dot missing at the end of the sentence.

Reviewer #2: The article presents a very interesting work, with a high number of patients about the impact of SVR obtained in the reduction of liver-death, but I have some issues that must be adressed:

1) The authors must describe very well in which manner they mantain the follow-up. Was only a clinical examination, or they performed an ultrasound, maybe an evluation of liver fibrosis and steatosis after SVR was obtaine?

2) The authors must explain if there is an impact of asociation of liver steatosis after SVR was obtained.

3) Liver steatosis was found to be an important factor that can cause advaned liver disease after SVR, due to the changes of beheaviour in patients (more food-intake, appereances of metabolic syndrome and T2DM) which can cause liver-trelated events and death in time. How was the impact of liver steatosis in those subjects, after SVR was obtained?

**Do you want your identity to be public for this peer review?** For information about this choice, including consent withdrawal, please see our Privacy Policy

Reviewer #1: No

Reviewer #2: **Yes: ** Robert Nastasa

---

## [Author Response · Author response to Decision Letter 1]

18 Jul 2025

Dear Editor

Thank you for the opportunity to re-submit our revised manuscript, “Effect of sustained viral clearance on liver-related mortality among individuals living with hepatitis C by treatment era: a population-based retrospective cohort study” (PONE-D-25-05248) for publication in the PLOS One.

We thank the Reviewers for their constructive feedback and insightful suggestions that helped us to improve the quality and clarity of our work. Please see our detailed response to each of the comments in the attached "Response to Reviewers" file. The changes are tracked in the revised manuscript. We are also submitting the clean copy, as requested.

We also noticed that Tables 1 and 2 were mistakenly duplicated in the original submission. This has been corrected, and the proper Table 1 has now been included.

Once again, we express our appreciation for your detailed review and consideration. Please kindly let us know if there are any further suggestions or comments.

Sincerely,

Beate Sander

---

## [Decision Letter · Decision Letter 1]

12 Aug 2025

Dear Dr. Sander,

Thank you for submitting your manuscript to PLOS ONE. After careful consideration, we feel that it has merit but does not fully meet PLOS ONE’s publication criteria as it currently stands. Therefore, we invite you to submit a revised version of the manuscript that addresses the points raised during the review process.

Your revised manuscript was reviewed by two original reviewers. Although one was satisfied with your modifications, the other pointed at a few remaining problems.  

We look forward to receiving your revised manuscript.

Kind regards,

Yury E Khudyakov, PhD

Academic Editor

PLOS ONE

Journal Requirements:

Reviewers' comments:

Reviewer's Responses to Questions

**Comments to the Author**

Reviewer #1: (No Response)

Reviewer #2: All comments have been addressed

2. Is the manuscript technically sound, and do the data support the conclusions?

Reviewer #1: Partly

Reviewer #2: Yes

3. Has the statistical analysis been performed appropriately and rigorously?

Reviewer #1: I Don't Know

Reviewer #2: Yes

4. Have the authors made all data underlying the findings in their manuscript fully available?

Reviewer #1: Yes

Reviewer #2: Yes

5. Is the manuscript presented in an intelligible fashion and written in standard English?

Reviewer #1: Yes

Reviewer #2: Yes

Reviewer #1: The authors improved the readabilty of abstract and introduction and corrected mistakes. The authors implied most of the reccomendations but forgot to correct SVR in the supplementary materials and I found some more flaws in the text.

These mistakes or inaccuracies I suggest correcting:

Please stick to HCV RNA positivity, it is a more proper term than RNA positivity (line 175, 185...).

Explain N in bellow the tables (it was deleted in some table descriptions (eg. table 2,3,4) but I think it stould stay there.

Throughout some tables HBsAG is written but both EASL and AASLD guidelines use abbreviation HBsAg (not capital G).

Throughout some tables abbreviation yrs. is used but not explained. Either explain it in the description of the table or write years as in the table 1.

Throughout some tables abbreviation ref. is used but not explained in the description of the tables.

Line 46: delete one bracket in )) = )

Line 104: formerly known or currently known as the Institute for Clinical Evaluative Sciences?

Line 201: DAA era should be DAA-era if you want to stick to the terminology with dash which is used further in the text

Line 205: vs - dot is missing - vs.

Line 233: space is missing after vs.

Line 243: dash is missing in pre-DAA-era

Line 276: CI: instead of C,I

Line 284+312: Time-to-liver-related mortality is not clear. There are too many dashes and I'm not sure if I understand the meaning of this term.

Line 309: Do you mean higher incidence of liver-related death?

Line 372: other word than realise would be more suitable

Line 393: What is the meaning of PHO HCV testing?

Line 414: liver-related death

Line 426: correct liver-realted death to liver-related death

Supplementary materials:

1. SVR is not corrected in the description of the abbreviations, it states sustained viral clearance.

2. All tables are named S1-S7 but bellow the tables there is written table A1-A7 instead of S1-S7. In the main text all the tables are named only above the table but bellow the tables there is only the description but not Table xxx and the description. Please unify this and correct.

Table S7: Sudy Cohort means study cohort?

Reviewer #2: The article has been improved with the recommendations that I made for the first manuscript, and now can be accepted for publication.

**Do you want your identity to be public for this peer review?** For information about this choice, including consent withdrawal, please see our Privacy Policy

Reviewer #1: No

Reviewer #2: No

---

## [Author Response · Author response to Decision Letter 2]

10 Sep 2025

The responses to the reviewers' comments are attached

---

## [Editor Report · Decision Letter 2]

16 Sep 2025

Effect of sustained virologic response on liver-related mortality among individuals living with hepatitis C by treatment era: a population-based retrospective cohort study

PONE-D-25-05248R2

Dear Dr. Sander,

We’re pleased to inform you that your manuscript has been judged scientifically suitable for publication and will be formally accepted for publication once it meets all outstanding technical requirements.

Kind regards,

Yury E Khudyakov, PhD

Academic Editor

PLOS ONE
---

## [Editor Report · Acceptance letter]

PONE-D-25-05248R2

PLOS ONE

Dear Dr. Sander,

I'm pleased to inform you that your manuscript has been deemed suitable for publication in PLOS ONE. Congratulations! Your manuscript is now being handed over to our production team.

Kind regards,

on behalf of

Dr. Yury E Khudyakov

Academic Editor

PLOS ONE